# A metallic molybdenum dioxide with high stability for surface enhanced Raman spectroscopy

Qiqi Zhang[1,2], Xinshi Li[1], Qiang Ma[1], Qing Zhang[1], Hua Bai[1], Wencai Yi[3], Jingyao Liu[3], Jing Han[1,4] & Guangcheng Xi[1,5]

Compared with noble metals, semiconductors with surface plasmon resonance effect are another type of SERS substrate materials. The main obstacles so far are that the semiconducting materials are often unstable and easy to be further oxidized or decomposed by laser irradiating or contacting with corrosive substances. Here, we report that metallic $MoO_2$ can be used as a SERS substrate to detect trace amounts of highly risk chemicals including bisphenol A (BPA), dichloropheno (DCP), pentachlorophenol (PCP) and so on. The minimum detectable concentration was $10^{-7}$ M and the maximum enhancement factor is up to $3.75 \times 10^6$. To the best of our knowledge, it may be the best among the metal oxides and even reaches or approaches to Au/Ag. The $MoO_2$ shows an unexpected high oxidation resistance, which can even withstand $300\,°C$ in air without further oxidation. The $MoO_2$ material also can resist long etching of strong acid and alkali.

[1] Institute of Industrial and Consumer Product Safety, Chinese Academy of Inspection and Quarantine (CAIQ), No. 11, Ronghua South Road, Beijing 100176, China. [2] Department of Chemistry, Capital Normal University, No. 105, North Road, West 3th Ring Road, Beijing 100048, China. [3] Laboratory of Theoretical and Computational Chemistry, Institute of Theoretical Chemistry, Jilin University, Changchun 130023, China. [4] Technical Test Center, Zhejiang Entry-Exit Inspection and Quarantine Bureau, No. 126, Fuchun Road, Hangzhou 310016, China. [5] Nanomaterials and Nanoproducts Inspection Research Center, General Administration of Quality Supervision, Inspection and Quarantine of the People's Republic of China (AQSIQ), No. 9, Madian East Road, Beijing 100088, China. Correspondence and requests for materials should be addressed to G.C.X. (email: xiguangcheng@caiq.gov.cn).

Surface-enhanced Raman spectroscopy (SERS) has become a powerful analytical tool in chemical, physical, biological sciences and so on[1–3]. Benefits from the rapid development of surface plasmon resonance (SPR) technology, detection of trace amounts of substances has been achieved by SERS, including pesticide and veterinary drug residues, environmental hormones, heavy metal ions and so on[4–6]. Different from normal Raman spectroscopy, SERS generally requires noble-metal nanocrystals with strong SPR effect as substrate materials[7,8]. The nature of the substrate material is one of the most critical factors to determine the performance of SERS[9,10]. An ideal SERS substrate material should include the following characteristics: strong SPR effect, high stability, low cost and good versatility[11]. So far, Au nanostructures are the most frequently used substrate materials in SERS due to their very strong SPR effects and highly chemical and thermal stability[12–15]. Ag nanocrystals are another widely studied SERS substrate material[16–18]. Although its price is much lower than that of Au, it is easy to be vulcanized by sulfur compounds in environment or oxidized by laser irradiation of Raman spectrometer, thus inevitably losing the SPR effect.

In addition to Au and Ag nanostructures, some semiconductor nanostructures with SPR effect, such as III–V semiconductor quantum dots[19], CuTe nanocrystals[20] and TiO$_2$ nanocrystals[21] have recently been reported to be used as active SERS substrate materials. However, a major obstacle in the practical application is that the electromagnetic enhancement factors (EFs) of these reported semiconductor materials are very low, normally within the range of 10–10$^3$, which is far less than the requirements of the detection of trace amounts of chemical and biological molecules. More recently, transition metal oxide nanostructures with high concentration of oxygen vacancy (such as TiO$_{2-x}$ and WO$_{2.83}$) have been shown to be promising for SERS substrate materials[22,23]. One outstanding example of this is the urchin-like W$_{18}$O$_{49}$ reported by Zhao et al., and its EF is even up to $3.4 \times 10^5$ level[24], which is known as the semiconducting material with the highest EF. Studies show that the strong SPR effects of the transition metal oxides result from their outer $d$-orbit free electrons induced by the oxygen vacancy contained in the crystal lattices[23]. Unfortunately, although the transition metal oxides have a much lower price compared to noble metals, their stability is very poor because these oxygen vacancies are easily removed by the high-temperature oxidation induced by the excitation light, normally provided by the laser beams with wavelengths from 500 to 700 nm of the Raman spectrometer. Once these oxygen vacancies are removed, the SPR effect of the material will disappear. For example, oxygen vacancies-rich W$_{18}$O$_{49}$ possesses the highest EF in the reported non-noble-metal SERS materials at present[24], but its SPR activity will be drastically reduced when it is exposed to air for several days even at room temperature[25,26]. Therefore, the discovery of robust SERS substrate materials with low cost and high stability is very meaningful both in basic research and practical applications.

As a common metal oxide, MoO$_2$ nanostructures are often used in the preparation of lithium ion batteries and electro-catalysts[27–30], but they are rarely reported for other uses. Compared with semiconducting MoO$_3$, MoO$_2$ has many vastly different characteristics, such as high conductivity, high melting point, high chemical stability and so on[31]. The results of the first-principles calculation show that MoO$_2$ presents a metallic character rather than semiconducting properties (Fig. 1), which is similar to the results of the previous theoretical calculations[27,32]. The highest occupied states of the MoO$_3$ are mostly composed of O$_{2p}$ orbitals, and the electrons are fully localized around the O atoms; but the region near the Fermi level of MoO$_2$ is composed of Mo$_{3d}$ orbitals, which presents the characteristic of the metal (Fig. 1a,b). At the same time, the free electron gas distribution, which was probed by calculating the electron localization functions (ELF), indicates that the free electron gas density of MoO$_2$ is far higher than that of MoO$_3$, and forms a lot of nonpolar Mo–Mo metallic bonds (Fig. 1c,d). Obviously, from MoO$_3$ to MoO$_2$, it has experienced a transition from a semiconductor to a conductor. Due to the existence of a large number of free electrons, MoO$_2$ is likely to have a strong SPR effect. If this conjecture is established, then combined with its high chemical stability, high melting point and low cost, MoO$_2$ is highly likely to be an ideal metal oxide-based SERS substrate material.

Herein, we report a new use of MoO$_2$, which can absorb visible light to produce strong SPR effect that resonate in the visible region. By using the SPR-active MoO$_2$ as SERS substrate, a series of high attention chemicals such as bisphenol A (BPA), dichloropheno (DCP) and pentachlorophenol (PCP) can be detected even at low level of $10^{-7}$ M and the maximum EF is up to $3.75 \times 10^6$. With regard to oxidation resistance, the MoO$_2$ shows an unexpected high stability, which can even withstand 300 °C of high-temperature heating in air without further oxidation. Furthermore, as a SERS substrate material, it also can resist the long time Laser irradiation and corrosion of strong acid and strong alkali. Combined with the low cost, the MoO$_2$ is promising as an active and universal SERS substrate material.

## Results

**Synthesis and characterizations.** The MoO$_2$ used in this study was synthesized by a simple hydrothermal method. Briefly, molybdenyl acetylacetonate (MA, 1 mmol) was to serve as molybdenum source was added under agitation to a mixture of ethanol (9 ml) and distilled water (41 ml). Then, the precursor solution was transferred to a Teflon-lined autoclave and sealed. After that, the above autoclave was slowly heated to 180 °C and kept at this temperature for 20 h. Finally, the as-obtained black products were washed with ethanol for three times. Figure 2a shows the schematic diagram of the whole synthesis process. As a metal oxide with intermediate valence, it should be noted that the synthetic reaction was carried out under relatively mild experimental conditions without the need for inert gas protection. The as-synthesized MoO$_2$ sample displays a positive temperature coefficient of resistance, and the obtained resistivity value is only $\sim 6.2 \times 10^{-3} \, \Omega \, cm$ at 300 K measured by a pressing plate method (Supplementary Fig. 1), suggesting it possesses a feature of electrical conductivity of metal as expected.

To obtain the accurate structure information, we first detect the crystal phase of the as-obtained product by powder X-ray diffraction (XRD). MoO$_2$ belongs to the structure of monoclinic type with the lattice parameters of $a = 5.6068$ Å, $b = 4.8595$ Å and $c = 5.5373$ Å. In this structure, O atoms are closely packed into octahedrons, and Mo atoms occupy half space of the octahedral void. The reverse edge-sharing MoO$_6$ octahedrons connect with each other to form a kind of deformed rutile structure (inset in Fig. 2b). Different from MoO$_3$, MoO$_2$ contains two kinds of Mo–Mo metallic bonds with different bond lengths (Mo–Mo distances were 0.25 and 0.31 nm, respectively), which makes it have good electrical conductivity. As shown in Fig. 2b, the XRD pattern of our product can be precisely indexed as the monoclinic-phase MoO$_2$ (JCPDS. 78-1069). No diffraction peaks of MoO$_3$ or other crystalline phases are found, suggesting that the as-obtained product is phase-pure MoO$_2$.

Furthermore, as another direct evidence, Raman spectroscopy was used to demonstrate that the sample is really monoclinic-phase MoO$_2$. As shown in Supplementary Fig. 2, main eight Raman scattering peaks at 200, 226, 345, 351, 456, 492, 569 and 739 cm$^{-1}$ are detected[30]. The characteristic peaks at 569 and

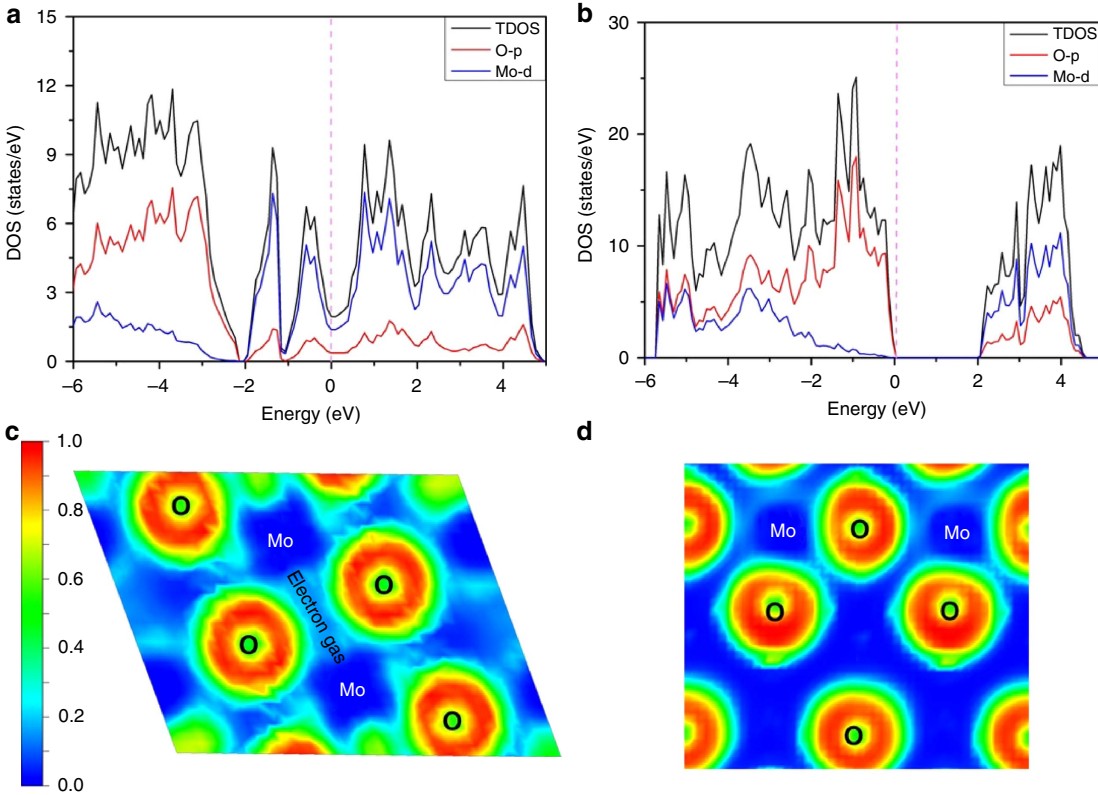

**Figure 1 | Electric structures of metallic MoO₂ and semiconducting MoO₃.** (**a,b**) Electronic density of states for $MoO_2$ and $MoO_3$, respectively. (**c,d**) The calculated ELF of $MoO_2$ and $MoO_3$, respectively. Green to red indicates the gradually increased charge localization.

$739\,cm^{-1}$ can be indexed to the O–Mo bond vibration modes of $MoO_2$, while the other fingerprint peaks at 200, 226, 345, 351, 456 and $492\,cm^{-1}$ can be attributed to the phonon vibration modes of $MoO_2$. In addition, Fourier transform infrared spectroscopy was used to prove that the surface of the sample has no residual organic matters introduced in the synthesis process (Supplementary Fig. 3). The bands at 500 and $780\,cm^{-1}$ can be attributed to the stretching vibrations of O–Mo units and the bridging oxygen atoms in O–Mo–O. The other bands at about 1,650, 3,480 and $2,330\,cm^{-1}$ can be well-attributed to the absorbed $H_2O$ and $CO_2$ molecules. The results confirmed that the surface of the obtained $MoO_2$ sample is considerably clean.

Then, the morphology and microstructure of the $MoO_2$ product were detected by transmission electron microscope (TEM) and scanning electron microscope (SEM). The low-magnification TEM image shown in Fig. 2c shows that the $MoO_2$ sample is composed of large quantity of dumbbell-like nanostructures. Interestingly, the enlarged TEM and SEM images (Fig. 2d,e and Supplementary Fig. 4) reveal that the dumbbell-like $MoO_2$ nanostructures are actually made up of many ultrathin nanosheets (2.5 nm in thickness) with sharp corners and edges, of which the geometric structure is very useful to the improvement of the SERS effect because such a hierarchical structure will produce a large number of high-density 'hot spots' (that is highly concentrated electromagnetic field) at nanoscaled gaps and sharp edges or corners[33–35]. The high-resolution TEM (HRTEM) image (Fig. 2f) and the corresponding fast Fourier transform pattern (Fig. 2g) demonstrated that the $MoO_2$ nanocrystals possess a high degree of crystallinity. The spacing of the lattice fringe of 0.48 and 0.24 nm can be indexed to the (101) and (111) planes of monoclinic $MoO_2$ (ref. 36), respectively. Energy-dispersive X-ray spectroscopy (EDS) suggested that the sample contains only two elements of Mo and O (Fig. 2h), and their ratio is very

close to 1:2. $N_2$ adsorption–desorption measurement revealed that the Brunauer–Emmett–Teller surface area of the $MoO_2$ nanodumbbells is $78.6\,m^2\,g^{-1}$ (Supplementary Fig. 5).

The valence states of Mo in the $MoO_2$ nanodumbbells were investigated by X-ray photoelectron spectroscopy (XPS). As shown in Fig. 3a, there are five obvious peaks in the survey spectrum of the $MoO_2$ nanodumbbells, which can be indexed to Mo3d (232.07 eV), C1s (283.1 eV), Mo3p (395.8 and 413.2 eV) and O1s (528.7 eV), respectively. Specifically, as shown in Fig. 3b, the typical four-peak-shaped Mo3d spectrum could be well fitted into two spin-orbit doublets, corresponding to $Mo^{4+}$ and $Mo^{6+}$ oxidation states, respectively[30]. The two characteristic strong peaks at 229.1 and 232.3 eV can be indexed to $Mo^{4+}$, while the other two weak shoulder peaks at 231.2 and 234.7 eV can be attributed to $Mo^{6+}$. According to the size of the peak areas, the concentration of $Mo^{4+}$ on the sample surface is much higher than that of $Mo^{6+}$, which clearly confirms that the molybdenum ion in the sample is basically tetravalent.

**Localized SPR effect and stability.** Ultraviolet–vis absorption spectrum shown in Fig. 3c clearly displayed that the $MoO_2$ nanodumbbells possess a considerable strong and well-defined visible absorption peak centred at 563 nm. For $MoO_2$, this interesting phenomenon is observed for the first time. Although the formation mechanism of this absorption band is not fully recognized, this optical behaviour is likely to be attributed to the SPR effect and believed to be closely related to its abundant d-orbit free electrons of the $MoO_2$. For comparison, because of the scarcity of the free electrons, $MoO_3$ nanodumbbells (Supplementary Fig. 6) obtained by oxidizing the $MoO_2$ nano-dumbbells at 600 °C in air does not exhibit this localized SPR effect: it has no absorption in the visible and near infrared

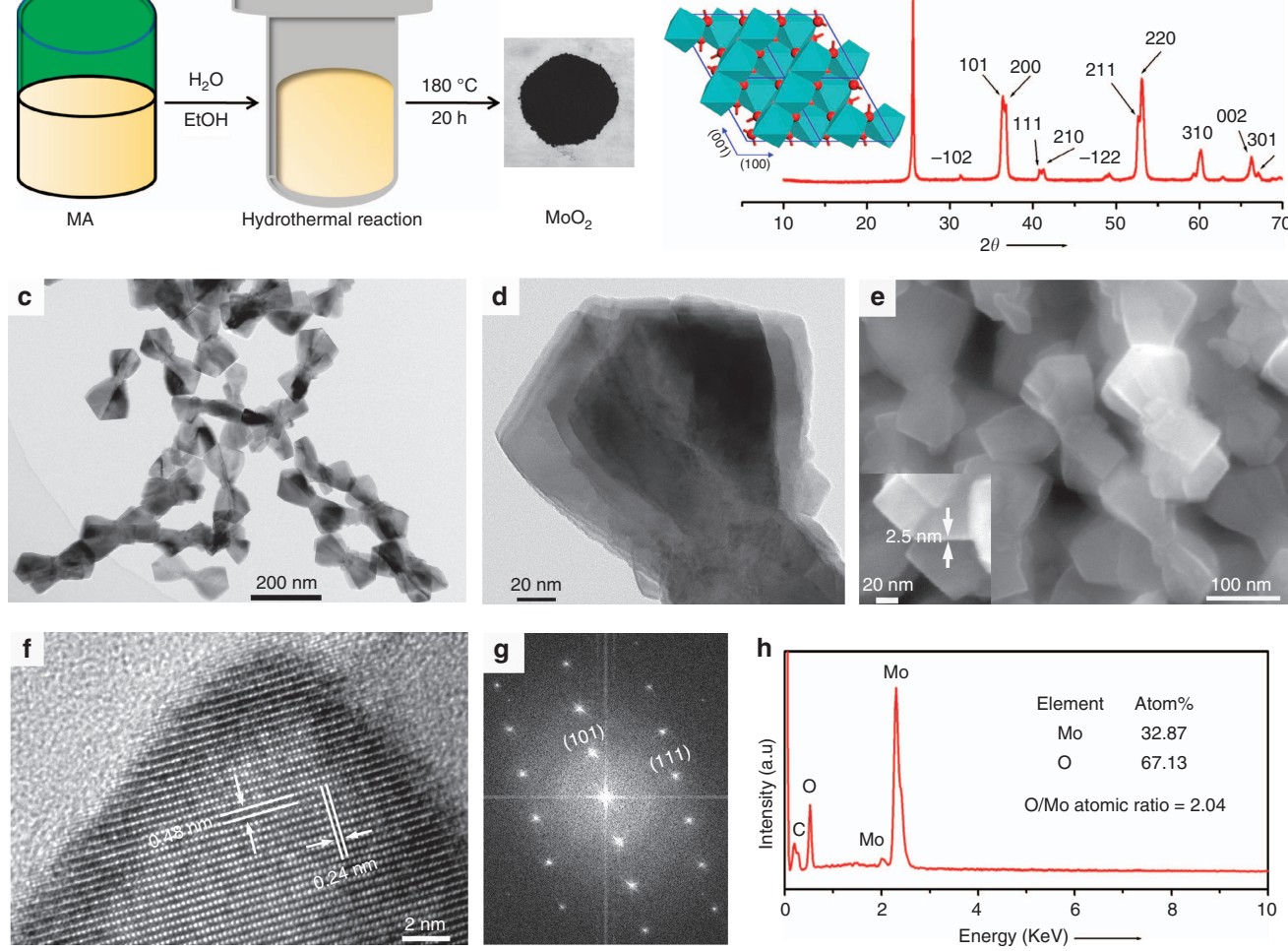

**Figure 2 | Synthesis, crystal structure, particle morphology and microstructure of MoO₂.** (**a**) Schematic illustrating the synthesis of the metallic MoO₂. (**b**) XRD pattern of the prepared MoO₂ powders. inset: crystal structure of monoclinic MoO₂. (**c,d**) TEM images of the MoO₂ powders. (**e**) SEM image of the MoO₂ sample. (**f,g**) HRTEM image and corresponding fast Fourier transform pattern of the MoO₂ particles. (**h**) EDS component analysis of the sample.

region (Supplementary Fig. 7). XPS measurement results reveal that only $Mo^{6+}$ ions were contained in the $MoO_3$ nanodumbbells (Supplementary Fig. 8), which further demonstrated that the strong localized SPR effect of the $MoO_2$ nanodumbbells results from the high concentration of free electrons of $Mo_{3d}$ orbitals. The $MoO_2$ nanodumbbells show the desirable optical properties, which highly consonant with the prediction.

Another surprising finding is that these $MoO_2$ nanodumbbells show unexpectedly high thermal and chemical stability compared with other SERS-active non-noble metal materials. For example, for the best known SERS-active metal oxide nanostructures, oxygen-deficient $W_{18}O_{49}$ nanowires (Supplementary Fig. 9) that have the highest EF ($3.4 \times 10^5$) in the reported metal oxides at present[24], comparative experiments have shown that their localized SPR peak quickly disappears due to the disappearance of the oxygen vacancies when slightly heated at 80 °C in air (Supplementary Fig. 10b); accordingly, their SERS activity completely disappeared. In stark contrast, these $MoO_2$ nanodumbbells still maintain remarkable localized SPR peak (Supplementary Fig. 10a), even after 300 °C of high-temperature heating for 24 h in air (Fig. 3d). The high thermal stability also demonstrates that the strong absorption band at 563 nm cannot be attributed to the charge transfer of ligand to $MoO_2$, because the most organic ligands cannot withstand 300 °C in air.

Furthermore, acid, alkali and photochemical stability test results demonstrated that the no detectable strength change was observed in the localized SPR peaks of the $MoO_2$ nanodumbbells (Fig. 3e,f). At the same time, a series of XPS spectra demonstrated that no detectable change in the surface valence states of Mo in the $MoO_2$ nanodumbbells after the heating, irradiating, acid and alkali corroding (Supplementary Fig. 11). Thus, the results demonstrated that the stability of these $MoO_2$ nanodumbbells is extraordinary high. In addition, compared with the more easily oxidized or corroded metal chalcogenides with localized SPR effect[37], the excellent stability of the as-obtained $MoO_2$ nanodumbbells is even more valuable.

**SERS properties of MoO₂ sample.** We use Rhodamine 6G (Rh6G), a common probe molecule, to examine the performance of these $MoO_2$ nanodumbbells as SERS substrate. Figure 4a shows the $MoO_2$-based SERS spectrum of Rh6G aqueous solution with a concentration of $10^{-6}$ M; its Raman scattering peaks are clearly visible, and all the peaks are in agreement with the Raman spectrum of the reference material of Rh6G (Supplementary Fig. 12). The commonest four characteristic peaks of Rh6G, $R_1$ ($612\,cm^{-1}$), $R_2$ ($773\,cm^{-1}$), $R_3$ ($1,363\,cm^{-1}$) and $R_4$ ($1,652\,cm^{-1}$) can be clearly observed.

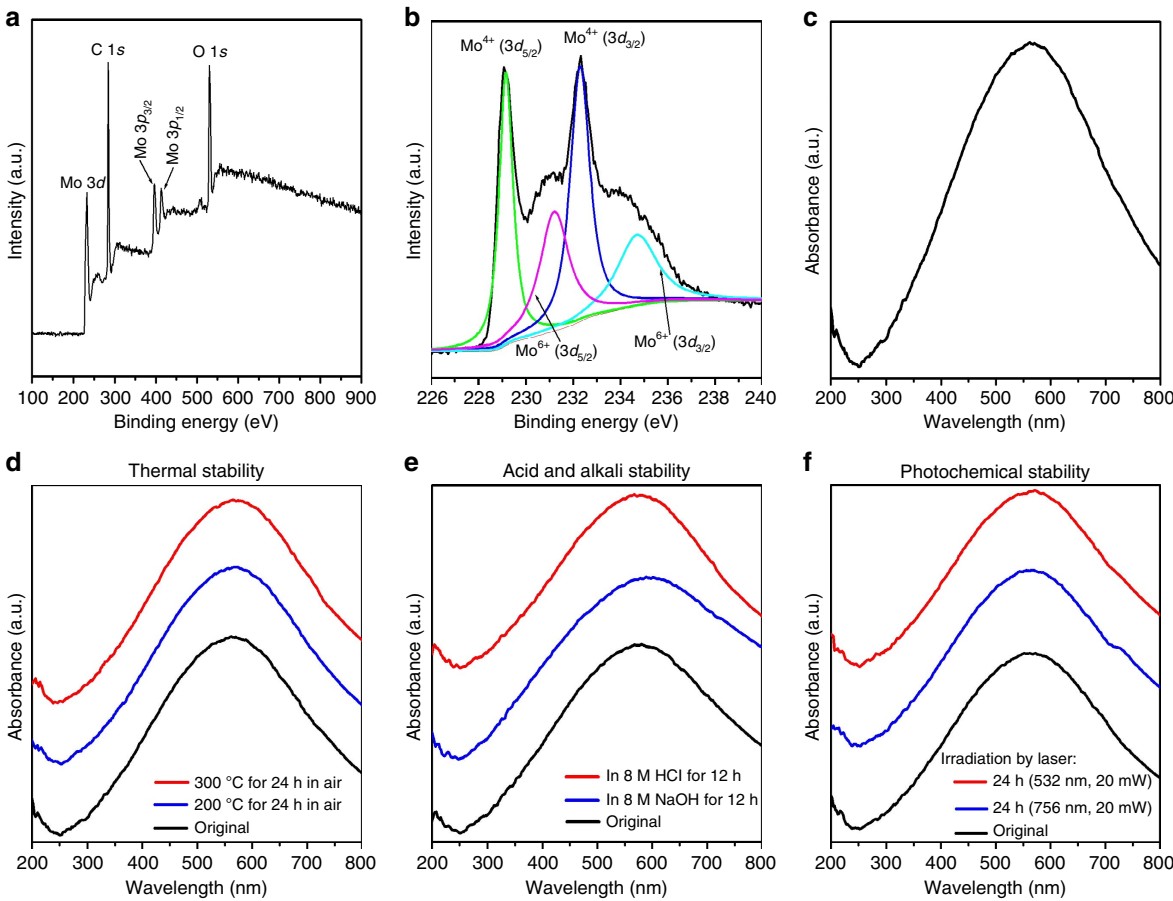

**Figure 3 | Valence states and ultraviolet–vis absorption characterizations of the MoO₂ nanodumbbells.** (**a,b**) XPS survey spectrum and Mo3*d* spectrum of the sample, which demonstrates that molybdenum ion in the sample is tetravalent. (**c**) Ultraviolet–vis absorption spectrum of the sample, showing a strong LSPR peak centred at 563 nm. (**d–f**) The LSPR peaks of these samples are almost the same after being heated in air (**d**), corroded with HCl and NaOH (**e**) and irradiated by laser (**f**), suggesting the high thermal and chemical stability of the MoO₂ nanodumbbells.

To distinguish whether the glass wafer played a contribution in the SERS since these MoO₂ samples were distributed on it, controlled experiments were carried out. The results indicated that no SERS spectra were obtained when bare glass wafer was used as the substrate (red spectrum in Fig. 4a), which definitely excludes the contribution of the glass wafer in the SERS measurements. On the other hand, when using the MoO₃ nanodumbbells without localized SPR effect as the substrate material, only its own Raman signals were detected and no SERS spectra of Rh6G were obtained (blue spectrum in Fig. 4a). These results demonstrated that the enhanced Raman signal really come from the MoO₂ nanodumbbells. Figure 4b shows the Raman spectra of four Rh6G samples with different concentrations from $10^{-4}$ to $10^{-7}$ M, indicating significant Raman enhancement in a wide concentration range and high detention sensitivity even at $10^{-7}$ M.

To verify whether these MoO₂ samples after high-temperature heating (300 °C in air) still have SERS activity, we have made a series of verification experiments, and found that these heated samples still showed excellent SERS activity for the detection of trace amounts of Rh6G (Fig. 4c), indicating the extremely high thermal stability of this material. For comparison, after only 80 °C of heating in air, the blue W₁₈O₄₉ nanowires (the reported metal oxide with the highest EF) was soon turned into yellow green and completely lost their SERS activity (Supplementary Fig. 13). These experimental results demonstrated that the MoO₂ nanodumbbells have broken through one of the biggest obstacles

in SERS applications: poor stability of the non-precious metal SERS substrates.

Subsequently, we used the Rh6G on bare glass and MoO₂ substrate to calculate the SERS EF of the MoO₂ nanodumbbells (Fig. 4d). The Raman scattering characteristic peaks (R₁ and R₂) of the Rh6G with three distinct concentrations ($10^{-4}$, $10^{-5}$ and $10^{-6}$ M) were measured. To ensure the accuracy of the results, the signal intensity of each characteristic peak at each concentration is averagely calculated from 50 measured points over the substrates. For characteristic peaks R₁, it can be seen that a series of tremendous EFs were obtained at each concentration. When the concentration was $10^{-6}$ M, the EF for R₁ even reached $3.75 \times 10^{6}$, which is about 10 times higher than that of the current highest EF recorded from W₁₈O₄₉ nanowires ($3.4 \times 10^{5}$). For R₂, although the obtained EFs were smaller than those obtained from R₁, the value also reached $10^{6}$ level. The results clearly demonstrated that the SERS enhancement effects of these MoO₂ nanodumbbells even can be comparable with that of noble metal nanostructures (Supplementary Table 1).

Based on the electromagnetic enhancement theory[1–3], the high EF of the MoO₂ nanodumbbells can be attributed to their strong SPR effect. The comparative experiments clearly prove this point. When the samples were heated at 350, 400, 450 °C for 5 h in air, with the increase of oxidation state and the reduction of free electron density, the plasma resonance absorption peak of them decreased violently. Accordingly, their corresponding SERS performance is also greatly decreased (Supplementary Fig. 14).

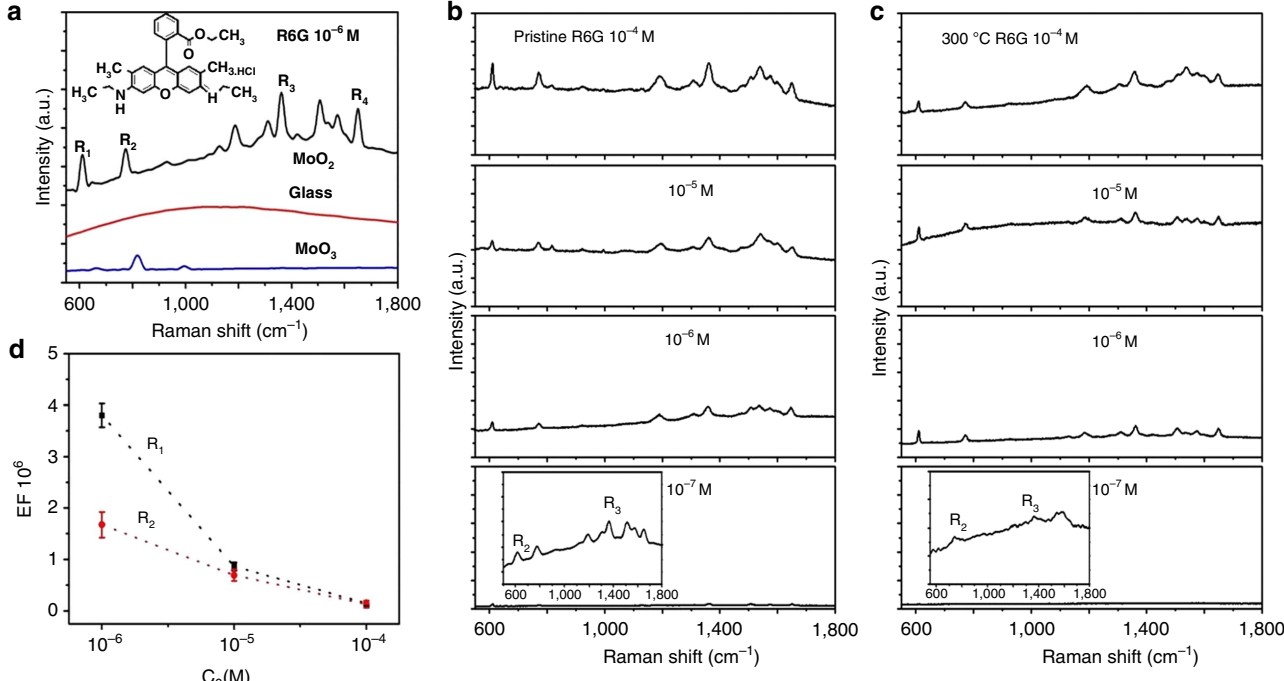

**Figure 4 | SERS measurements of Rh6G with the as-prepared MoO$_2$ nanodumbbells. (a)** Raman spectra of $10^{-6}$ M Rh6G aqueous solution obtained in MoO$_2$ nanodumbbells, bare glass and MoO$_3$ nanodumbbells. **(b)** Gradually weakened Raman scattering signals recorded from Rh6G aqueous solution at four different concentration levels ($10^{-4}$, $10^{-5}$, $10^{-6}$, $10^{-7}$ M), suggesting the MoO$_2$ nanodumbbells have greatly enhanced Raman scattering, with a low detection limit of $10^{-7}$ M. **(c)** These MoO$_2$ nanodumbbells still has high Raman enhancement effects even after 300 °C of high-temperature heating in air. **(d)** The average Raman EFs obtained by counting the peak intensities (R$_1$ and R$_2$) at three different concentration levels.

On the other hand, the effect of charge transfer also plays an important role in improving the EF. As a direct evidence, comparative experiments have shown that the ultraviolet−vis absorption of the R6G-modified MoO$_2$ nanodumbbells showed several new absorption bands at 349, 485, 526, 580 and 732 nm when compared to the ultraviolet−vis spectrum of unmodified-MoO$_2$ nanodumbbells (Supplementary Fig. 15). These experimental phenomena clearly indicated that there is a distinctly charge-transfer between MoO$_2$ and R6G, and the electrons transfer direction is from the MoO$_2$ nanodumbbells to the R6G molecules based on the direction of spectral shifts[38]. Furthermore, it should be noted that the peaks at 612 and 773 cm$^{-1}$ are well-known to be vibronically coupled[39], which are really among the most enhanced peaks in the SERS spectra. These results are strong indications of charge-transfer contributions to the SERS[40].

For practical SERS applications, in addition to high sensitivity, reliable reproducibility is another important factor. To demonstrate that these MoO$_2$ nanodumbbells have high reproducibility, SERS signal detection was executed by using Rh6G as probe molecule ($10^{-7}$ M). Figure 5a shows the optical photograph of a randomly selected area (70 µm × 70 µm) of the as-fabricated SERS substrate, indicating the uniform distribution of the MoO$_2$ nanodumbbells. In this area, 100 randomly selected points were used for SERS detection, and the results show that the obtained SERS signals are highly similar (Fig. 5b). To more fully confirm the reproducibility of the MoO$_2$ nanodumbbells, SERS spectra of 5,030 randomly chosen measurement points in this area were used to calculate their relative s.d. (RSD). The SERS mapping of the 5,030 measurement points is shown in Fig. 5c. The intensities of the characteristic peak R$_1$ at 612 cm$^{-1}$ obtained from the 5,030 sets of data of the SERS mapping are shown in Fig. 5d. By using Bessel formula, the RSD of these measured intensities was calculated to be only about 4.7%. Furthermore, it

was calculated that the RSD of the characteristic peak (R$_2$) intensities at 773 cm$^{-1}$ is only about 5.2% (Fig. 5e,f). These experimental results confirm that high reproducibility can be achieved in one piece of MoO$_2$ substrate. Then, for different batches of MoO$_2$ substrates, how about the reproducibility of them? To figure out this problem, 32 pieces of MoO$_2$ substrates were fabricated, and the intensities of the characteristic peak R$_1$ were measured from five points randomly selected in every piece. The calculated average RSD are 4.9%, 8.1%, 10.8% and 10.5% for Rh6G at $10^{-4}$, $10^{-5}$, $10^{-6}$ and $10^{-7}$ M, respectively (Supplementary Fig. 16). We also followed the same steps to measure the intensities of characteristic peak R$_2$ at 775 cm$^{-2}$, and the calculated average RSD are 7.1%, 11.6%, 13.9% and 10.9% for Rh6G at $10^{-4}$, $10^{-5}$, $10^{-6}$ and $10^{-7}$ M, respectively (Supplementary Fig. 17). These results clearly demonstrated that the MoO$_2$ substrate possesses excellent reproducibility.

Further investigation demonstrated that these MoO$_2$ nanodumbbells have a good universality for trace chemical detection as SERS substrate. Specifically, in addition to Rh6G, other common azo dyes, such as rhodamine B (RhB), methyl orange, methyl blue and fuchsin acid, can also be determined even at an extremely low concentration of $10^{-7}$ M (Supplementary Fig. 18). More importantly, as a practical application, the present MoO$_2$-based SERS technology can be used to accurately detect trace level polyphenols and polychlorinated phenols which are highly concerned environmental hormones. As a polyphenol compound, BPA, also known as plasticizer is a chemical that seriously affects the metabolism of hormones in animals, and many countries have listed it as a prohibited substance. However, due to some technical reasons, a lot of common consumer goods, such as children's toys, food packaging materials and cosmetics are still being contaminated by a small amount of BPA. Aiming at the detection of BPA, as shown in Supplementary Fig. 19,

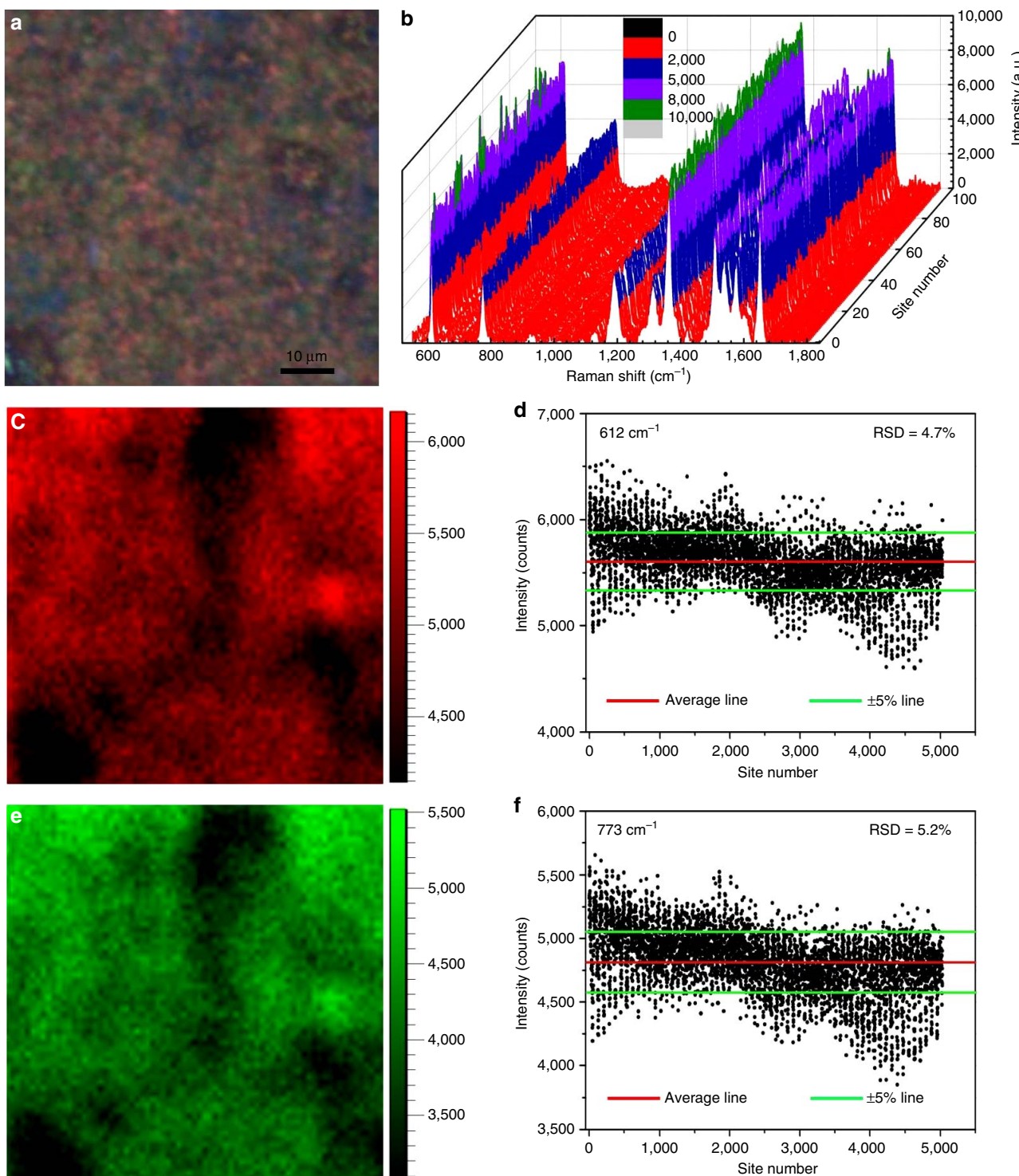

**Figure 5 | Determination of the signal reproducibility and uniformity of the MoO₂ substrate.** (**a**) Optical photograph of the substrate covered with MoO₂ nanodumbbells. (**b**) SERS signals collected from 100 randomly selected points on the substrate. (**c,d**) The SERS mapping and signal intensities at 612 cm⁻¹ of 10⁻⁷ M Rh6G in the region shown in **a**. (**e,f**) The SERS mapping and signal intensities at 773 cm⁻¹ of 10⁻⁷ M Rh6G in the region shown in **a**.

$10^{-4}$–$10^{-7}$ M BPA can be detected by using the MoO₂-based SERS method. Polychlorinated phenols, such as 2,4-dichlorophenol (2,4-DCP), 2,4,5-trichlorophenol (2,4,5-TCP), 2,3,4,6-tetrachlorophenol (2,3,4,6-TeCP) and PCP are another kind of chemical substance which is highly concerned environmental hormones. Fortunately, such compounds can also be detected by this MoO₂-based SERS method. Figure 6 and Supplementary Figure 20 show the SERS spectra of the 2,4-DCP, 2,4,5-TCP,

2,3,4,6-TeCP and PCP. These results clearly confirmed that the MoO₂ nanocrystals are possessed of superior applicability and generality as a SERS substrate material.

## Discussion

In summary, MoO₂ nanodumbbells with sharp corners and nanoscaled gaps have been synthesized by a simple and

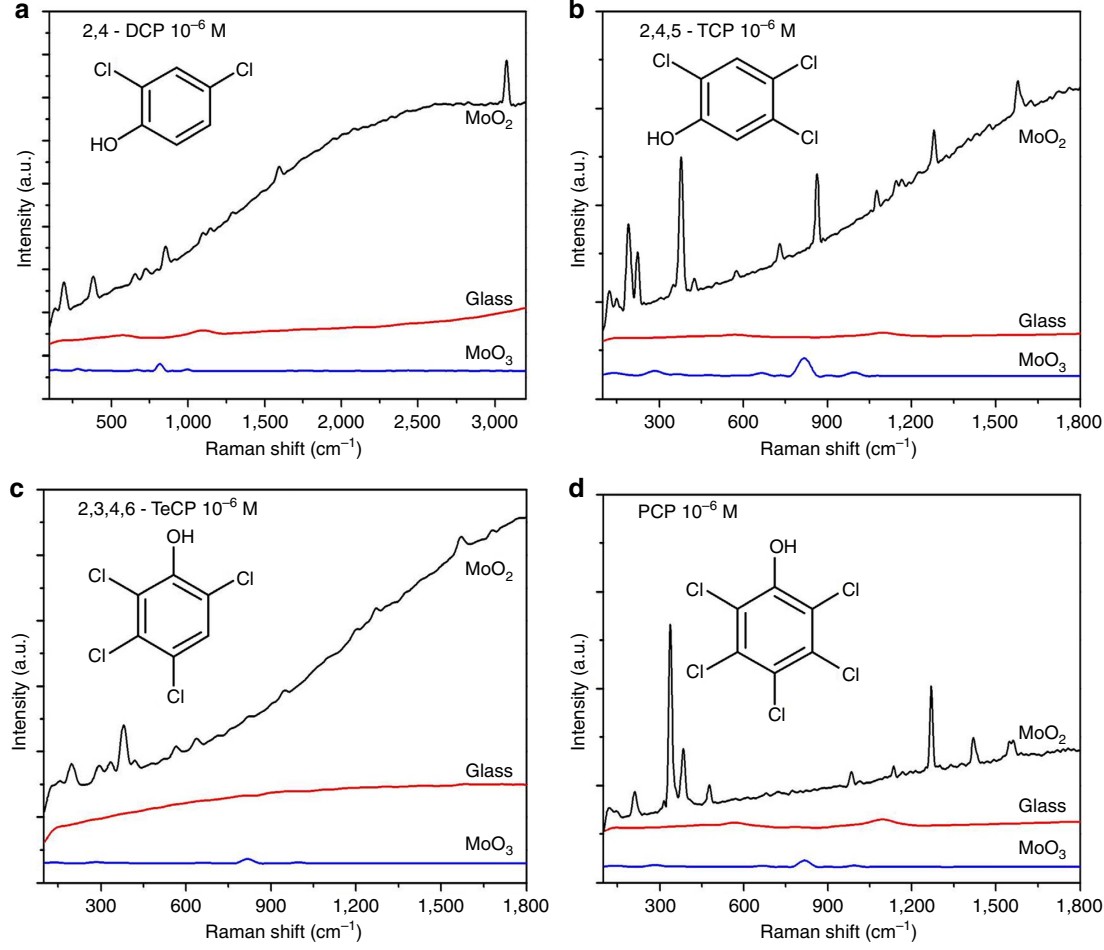

**Figure 6 | SERS spectra of a series of Polychlorinated phenols (PCPs).** (**a**) 2,4-DCP. (**b**) 2,4,5-TCP. (**c**) 2,3,4,6-TeCP. (**d**) PCP.

surfactant-free hydrothermal method. These $MoO_2$ nanodumb-bells contain high concentration of free electrons and low resistivity, which make them have a strong and well-defined SPR property. Compared with other reported SPR-active non-noble metal nanocrystals, these $MoO_2$ nanodumbbells show an extremely impressive thermal and chemical stability, which even can endure 300 °C of heating in air without being oxidized. The remarkable stability ensures that their SPR characteristics will not disappear when irradiated by laser or contact corrosive substances. These properties endow $MoO_2$ with a new use: as a non-noble metal substrate for SERS, the $MoO_2$ nanocrystals can be used to detect a series of highly risk compounds, such as BPA, DCP, TCP, PCP and so on. The results of this research have broken through an obstacles in the application of metal oxides in SERS for a long time, which really realized the preparation and applications of sensitive and universal non-noble metal-based SERS substrate materials with high stability.

## Methods

**Synthesis of $MoO_2$ nanodumbbells.** All chemicals used in the experiments are of analytical purity. In a typical synthesis, 0.1 g of molybdenyl acetylacetonate ($[CH_3COCH=C(O)CH_3]_2MoO_2$) was added into a mixed solution of distilled water (41 ml) and absolute ethanol (9 ml), and stirred for an hour at room temperature. And then, add the mixture into a Teflon-lined stainless steel autoclave and heat it for 20 h at 180 °C. After the reaction is completed, the black products were separated and collected by high speed centrifugation. Finally, the black powders were washed with ethanol and distilled water for three times and dried at 50 °C in a vacuum drying oven.

**Characterization.** These samples were measured by a variety of characterization techniques. XRD patterns of the products were obtained on a Bruker D8 focus X-ray diffractometer by using CuKα radiation ($\lambda = 1.54178$ Å). SEM images and EDS were obtained on a Hitachi S-4800. TEM and HRTEM characterizations were performed on a Tecnai G F30 operated at 300 kV. Ultraviolet–vis absorption spectra were recorded with a Shimadzu UV3600. XPS experiments were performed in a Theta probe (Thermo Fisher) using monochromated Al Kα X-rays at $hv = 1486.6$ eV. Peak positions were internally referenced to the C1s peak at 284.6 eV. The Fourier transform infrared spectra were measured from THERMO Iz-10. The specific surface area was measured in a Micro Tristar II 3020. XPS were recorded on an ESCALab-250Xi of ThermoFisher Scientific.

**Raman tests.** To study the SERS of these $MoO_2$ nanodumbbells, a confocal micro Raman spectrometer (Renishaw, inVia) is used as the measuring instrument. In all SERS tests, the excitation wavelength is 532.8 nm, laser power is 0.5 mW and the specification of the objective is × 50 L. A series of standard solution (aqueous) of highly risk chemical with concentrations of $10^{-4}$–$10^{-7}$ M were used as the probe molecules. To improve the signal reproducibility and uniformity, the $MoO_2$ nanodumbbells were dipped into a probe solution to be measured for 20 min, then taken out and dried in air for 1 h. In all SERS tests, the laser beam is perpendicular to the top of the sample to be tested with a resultant beam spot diameter of 5 μm. The calculation of EF are provided in Supplementary Methods.

**Electronic structure calculations.** All density functional theory calculations and ELF were carried out using the Vienna *abinitio* simulation package. Details of the calculations are provided in Supplementary Methods.

**Data availability.** The data that support the findings of this study are available from the corresponding author on reasonable request.

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

## Acknowledgements

This work received financial support from the Dean Fund of Chinese Academy of Inspection and Quarantine (2016JK025), the Science Foundation of AQSIQ (2015IK308) and the Natural Science Foundation of China (51472226, 21373098).

## Author contributions

G.X. proposed and designed the project; and Qiq.Z. and X.L. prepared MoO$_2$ materials. Q.M. characterized ultraviolet–vis and XRS measurement; Qin.Z. and H.B. performed SEM and TEM characterization; Qiq.Z. conducted XRD, Raman, FTIR, SERS and EFs measurement. W.Y. and J.L. performed electronic structure calculations. J.H. performed FTIR and conductivity characterization. All authors critically evaluated the manuscript.

## Additional information

**Competing interests:** The authors declare no competing financial interests.

