## [Peer Review File · Nature Communications]

Reviewers' comments:

Reviewer #1 (Remarks to the Author):

This is a very interesting article and should be published. The authors demonstrate a method to produce metallic MoO₂ nano-dumbbells as SERS substrate and demonstrate that they produce high enhancements. They show that there is a surface plasmon resonance in the visible region of the spectrum, and that the resistivity is low indicating metallic nature and they back this up with DFT calculations.

There are a few problems with the manuscript which need attention. Their claim that W18O49 has the highest enhancement observed for semiconductors is incorrect. It is only true for oxides. More generally ZnSe for example has a higher enhancement factor. There are several English errors, such as on page 8, six lines from the bottom, they mean "quickly" not "quick". On the next line remove "was" after "activity".

The major problem with this manuscript is in the interpretation of their results. They totally ignore the possibility of charge-transfer contributions, despite the fact that the relatively large enhancements for their main test molecule (R6G) often has considerable such contributions. Note that the 612 cm⁻¹ line and the 773cm⁻¹ line are well-known to be vibronically coupled (J. Phys. Chem. 1984, 24, 5935). These are among the most enhanced lines in their spectrum, and that is a strong indication of charge-transfer contributions to the SERS effect (J. Phys. Chem. C 2008, 112, 5605). The authors should include this possibility in their discussion.

Otherwise this has the potential to be a ground-breaking article, considering the practical problems using Ag or Au as SERS substrates. These metals are subject to instability and degradation, while it appears the MoO₂ substrates presented here have the possibility to transform the field of SERS if they work out. I am planning to try it in my laboratory.

Reviewer #2 (Remarks to the Author):

This paper details the formation of a new SERS substrate using molybdenum dioxide. The work is well carried out- however in my opinion it is not of sufficient novelty for the journal. One of the key selling points of the paper is the expense and difficulty of using gold for detection purposes. This is not true in my opinion- only a very tiny fraction of gold is used for SERS and this really does not cost much.

The manuscript uses nanodumbbells throughout. However I don't think that is an accurate description of the particles. Further I am not sure what link the shape of the particle has on the actual performance. Further the paper makes reference to oxygen vacancies throughout as a means of justifying the results- however this is not quantified or measured.

The use of diagrams is good and the enhancement factors seen are reasonable but not in the league of gold or silver nanoparticles.

I think the work would be suitable for publication in advanced functional materials and would suggest that the authors submit a revised manuscript to that journal.

Reply for the first reviewer

1. Their claim that $W_{18}O_{49}$ has the highest enhancement observed for semiconductors is incorrect. It is only true for oxides. More generally ZnSe for example has a higher enhancement factor.

Reply: we thank the reviewer for this suggestion and we are sorry for our carelessness. According to the suggestion of the reviewer, the related description has been corrected in the revised manuscript (second line, second page). As for ZnSe, although it has a higher enhancement factor, it is well known that its thermal stability and acid and alkali stability are very poor. So it is not suitable for the actual SERS detection.

2. There are several English errors, such as on page 8, six lines from the bottom, they mean “quickly” not “quick”. On the next line remove “was” after “activity.

Reply: we are sorry for our carelessness. The manuscript has been carefully checked and some English errors have been corrected. We are willing to take this paper to the professional polish companies if it's necessary.

3. They totally ignore the possibility of charge-transfer contributions, despite the fact that the relatively large enhancements for their main test molecule (R6G) often has considerable such contributions. Note that the 612 cm^{-1} line and the 773 cm^{-1} line are well-known to be vibronically coupled (J. Phys. Chem. 1984, 24, 5935). These are among the most enhanced lines in their spectrum, and that is a strong indication of charge-transfer contributions to the SERS effect (J. Phys. Chem. C 2008, 112, 5605). The authors should include this possibility in their discussion.

Reply: we are grateful for the constructive suggestion given by the reviewer. We have done some experiments (see Supplementary Figure 14) according to the suggestions of the reviewer, and the discussion on the mechanism of charge-transfer has been added in the revised paper (see third paragraph, page 10):

On the one hand, the high EF of the MoO_2 nanodumbbells can be attributed to their strong SPR effect. On the other hand, the effect of charge-transfer also plays an important role in improving the EF. As a direct evidence, comparative experiments

shown that the UV–vis absorption of the R6G-modified MoO₂ nanodumbbells appeared several new absorption bands at 349, 485, 526, 580, and 732 nm when compared to the UV–Vis spectrum of unmodified MoO₂ nanodumbbells (Supplementary Figure 14). These experimental phenomena clearly indicated that there is a distinctly charge-transfer between MoO₂ and R6G, and the electrons transfer direction is from the MoO₂ nanodumbbells to the R6G molecules based on the direction of spectral shifts³⁹. Furthermore, it should be noted that the peaks at 612 cm⁻¹ and 773 cm⁻¹ are well-known to be vibronically coupled⁴⁰, which are really among the most enhanced peaks in the SERS spectra. These results are strong indications of charge-transfer contributions to the SERS⁴¹.

References

39. Joy, V. T. & Srinivasan, T. K. K. Fourier-transform surface-enhanced Raman scattering study on thiourea and some substituted thioureas adsorbed on chemically deposited silver films. *Spectrochem. Acta. A* **55**, 2899—2909 (1999).
40. Hildebrandt, P. & Stockburger, M. Surface-enhanced resonance Raman spectroscopy of Rhodamine 6G adsorbed on colloidal silver. *J. Phys. Chem.* **88**, 5935-5944 (1984).
41. Lombardi, J. R. & Birke, R. L. A. Unified approach to surface-enhanced Raman spectroscopy. *J. Phys. Chem. C* **112**, 5605-5617 (2008).

Reply for the first reviewer

1. The work is well carried out- however in my opinion it is not of sufficient novelty for the journal. One of the key selling points of the paper is the expense and difficulty of using gold for detection purposes. This is not true in my opinion- only a very tiny fraction of gold is used for SERS and this really does not cost much.

Reply: If only from the point of view of Raman spectroscopy studies, the required gold really is not too much in the experiments. However, for actual testing requirements, SERS technology based on the Au nanostructures is difficult to be extended to large-scale detections, such as screening of hazardous substances in

consumer goods and entry-exit inspection and quarantine.

Cheap price is only one of the selling points of our work. Compared with traditional noble metals with high enhancement factors, semiconductors with localized SPR effects are another type of SERS substrate materials. Alarming, the main obstacles in applications so far are that the semiconducting materials exhibit in general poor stability. This raises an urgent problem whether there is an inexpensive non-noble metal material can achieve a high EF ($> 10^6$), while maintaining long-term stability. Our research focuses on solving these urgent issues. This allowed for the first time the find of a new use of metallic MoO₂, which be used as a general SERS substrate to detect a series of trace amounts of highly risk chemicals, whose EF is up to 3.75×10^6 . Such a high EF creates a new record among the metal oxide SERS materials. More importantly, the metallic MoO₂ shows a unexpectedly high thermal and chemical stability compared with other SERS-active non-noble metal materials. which can even withstand 300 °C of high temperature in air without further oxidation. The MoO₂ material also can withstand the long chemical etching process of strong acid and strong alkali. *Just like what the said of the first reviewer: this has the potential to be a ground-breaking article.*

2. The manuscript uses nanodumbells throughout. However I don't think that is an accurate description of the particles. Further I am not sure what link the shape of the particle has on the actual performance.

Reply: we thank the reviewer for this suggestion. We also hope to be able to use a word to describe its morphology accurately. In the shapes of all we can think of, such as dumbbell-like, bow tie-like, and spindle-like, dumbbell-like may be a more accurate description.

Interestingly, the enlarged TEM and SEM images (Fig. 2d-e and Supplementary Fig. 4) reveals that the dumbbell-like MoO₂ nanostructures are actually made up of many ultrathin nanosheets (2.5 nm in thickness) with sharp corners and edges, which geometric structure is very useful to the improvement of the SERS effect because such a hierarchical structure will produce a large number of high-density "hot spots"

(that is highly concentrated electromagnetic field) at nanoscaled gaps and sharp edges or corners³⁴⁻³⁶.

3. Further the paper makes reference to oxygen vacancies throughout as a means of justifying the results- however this is not quantified or measured.

Reply: we thank the reviewer for this suggestion, however, it should be noted that we don't quite understand the question because we do not take the oxygen vacancies as a mechanism to explain our results. The oxygen vacancies mentioned in the article refer to those oxygen vacancies-rich semiconductors, such as $W_{18}O_{49}$. We choose oxygen vacancies-rich $W_{18}O_{49}$ as reference material is to show that although it has a very high SERS enhancement factor, but its thermal and chemical stability is very poor, and it is not suitable as the actual SERS substrate material. As a sharp contrast, one of the highlights of this work is that the metallic MoO_2 not only has a high enhancement factor, but also has a very high thermal and chemical stability, which is precisely what the semiconductor cannot be achieved.

4. The use of diagrams is good and the enhancement factors seen are reasonable but not in the league of gold or silver nanoparticles.

Reply: we thank the reviewer for this suggestion. In the early days, the enhancement factor of gold and silver is about 10^5 - 10^6 level, then, along with the development of nanotechnology, a variety of unique gold and silver nanostructures were prepared by variety of control systems, the enhancement factor of gold and silver increases to 10^6 - 10^8 level. Therefore, in the text (Third lines, page 2), our presentation on the properties of these metallic MoO_2 is "reaches or approaches to Au/Ag".

Reviewers' comments:

Reviewer #1 (Remarks to the Author):

The authors have addressed my concerns about the manuscript and I think it is now ready for publication. Let me reiterate that this work has the potential to be transformative in the SERS community if it can easily be reproduced by other workers.

Reviewer #2 (Remarks to the Author):

I do not feel that the authors have adequately addressed my points. The cost for using gold is negligible in these devices, i do think the existing gold base materials could be scaled.

the authors do not provide any mechanistic insight into the SERS enhancement- this is what in my mind really differentiates a nature comms paper - covering the "why" question not a "what" question. This is still lacking in the work. If the authors can further revise their manuscript and give a description of why they get an enhancement it would be worth publishing here. otherwise i suggest AFM.

Reply for the reviewer

1. I do not feel that the authors have adequately addressed my points. The cost for using gold is negligible in these devices, i do think the existing gold base materials could be scaled.

Reply: we thank the reviewer for this suggestion. According to the suggestion of the reviewer, the related description has been changed into "As an active SERS substrate material, although the consumption of gold is very little or even negligible, it will further reduce the using cost and accelerate the large-scale detections, such as screening of hazardous substances in consumer goods and entry-exit inspection and quarantine if we can find a stable SERS substrate material that is cheaper than gold" (revised manuscript: second paragraph, second page).

2. The authors do not provide any mechanistic insight into the SERS enhancement- this is what in my mind really differentiates a nature comms paper-covering the "why" question not a "what" question. This is still lacking in the work. If the authors can further revise their manuscript and give a description of why they get an enhancement it would be worth publishing here. otherwise i suggest AFM.

Reply: we thank the reviewer for this suggestion. Generally, the two key contributions to the SERS enhancement are the electromagnetic (EM) factor and the chemical contribution¹⁻⁴. The EM enhancement of SERS-active substrates is by far the most important, and it is mainly determined by the nanostructure of the metallic surface and the wavelength-dependent dielectric properties of the materials. The conduction electrons of these nanoscale features can be driven by the incident electric field in collective oscillations known as localized surface plasmon resonances (SPRs). We believe that the enhancement mechanism of the SERS is mainly derived from the strong SPR effect of metallic MoO₂. To prove it, we first theoretically proved that the MoO₂ has a strong metallic property and high free-electron density (please see second paragraph, fourth page). Due to the existence of a large number of free electrons, the metallic MoO₂ is likely to have a strong SPR effect. Furthermore, the as-synthesized

MoO₂ sample displays a positive temperature coefficient of resistance (PTC), and the obtained resistivity value is only $\sim 6.2 \times 10^{-3} \Omega \text{ cm}$ at 300 K measured by a pressing plate method (please see second paragraph, fifth page), suggesting it possesses a feature of electrical conductivity of metal as expected. Then, the UV-Vis absorption spectrum shown in Figure 3c clearly displayed that the MoO₂ nanodumbbells possess a considerable strong and well-defined visible absorption peak centered at 563 nm (please see first paragraph, eighth page). Finally, we demonstrated that the SPR-active MoO₂ have excellent SERS performance. Therefore, all the evidence has shown that the enhancement mechanism originates from the strong SPR effect of MoO₂, that is the EM enhancement.

On the basis of the above analysis, in order to further demonstrate this mechanism, more experiments were done in the revised paper according to the suggestion of the reviewer. When the samples were heated at 350, 400, 450 °C for 1 h in air, with the increase of oxidation state and free electron density decreased, the plasma resonance absorption peak of them decreased violently. Accordingly, their corresponding SERS performance is also greatly decreased (revised manuscript: third paragraph, tenth page).

REVIEWERS' COMMENTS:

Reviewer #2 (Remarks to the Author):

This paper still needs some further work but is better than the previous two versions i have seen. I suggest it is excepted after some further work.

Firstly i would remove the argument about cost of Au nano particles. This has been toned down but really does not add much.

I am really not certain about the band at ca 550nm as being an SPR band - is there a definitive proof for this. Cant it just be a ligand to metal charge transfer band - or a d-d transition? The evidence for a related SPR band is not really present?

Response to The Referees

1. Firstly I would remove the argument about cost of Au nanoparticles. This has been toned down but really does not add much.

Reply: We now agree with the reviewer's opinion, and the discussion on the cost of gold has been completely removed from the text.

2. I am really not certain about the band at ca 550nm as being an SPR band- is there a definitive proof for this. Cant it just be a ligand to metal charge transfer band- or a d-d transition? The evidence for a related SPR band is not really present?

Reply: we thank the reviewer for this suggestion. We agree with the reviewer's opinion: it is really not certain about the band at 563 nm as being an SPR band. However, there are some evidences that the band at 563 nm is most likely to be the SPR band. Firstly, theoretical calculation and simulation proved that the MoO₂ has a strong metallic property and high free-electron density. Secondly, the as-synthesized MoO₂ sample displays a positive temperature coefficient of resistance (PTC), and the obtained resistivity value is only $\sim 6.2 \times 10^{-3} \Omega \text{ cm}$ at 300 K measured by a pressing plate method, suggesting it possesses a feature of electrical conductivity of metal as expected. These two features provide necessary prerequisites for the generation of the SPR behavior. Thirdly, thermal, acid, alkali, and photochemical stability test results shown that the no detectable strength change was observed in the localized-SPR peaks of the MoO₂ samples, which demonstrates that the band at 563 nm can't be attributed to the charge transfer of ligand to MoO₂, because the ligands cannot withstand 300 °C in air or the corrosion of strong acid and alkali. Of course, the accurate formation mechanism of the band at 563 nm still needs further exploration. The discussion please see page 8 and page 9.